# Sulfonamide Porphyrins as Potent Photosensitizers against Multidrug-Resistant *Staphylococcus aureus* (MRSA): The Role of Co-Adjuvants

**DOI:** 10.3390/molecules28052067

**Published:** 2023-02-22

**Authors:** Sofia N. Sarabando, Cristina J. Dias, Cátia Vieira, Maria Bartolomeu, Maria G. P. M. S. Neves, Adelaide Almeida, Carlos J. P. Monteiro, Maria Amparo F. Faustino

**Affiliations:** 1LAQV-Requimte and Department of Chemistry, University of Aveiro, 3810-193 Aveiro, Portugal; 2CESAM, Department of Biology, University of Aveiro, 3810-193 Aveiro, Portugal

**Keywords:** porphyrins, sulfonamides, MRSA, photodynamic therapy, antimicrobial resistance, photosensitizer, singlet oxygen, potassium iodide, gram-positive bacteria, *Staphylococcus aureus*

## Abstract

Sulfonamides are a conventional class of antibiotics that are well-suited to combat infections. However, their overuse leads to antimicrobial resistance. Porphyrins and analogs have demonstrated excellent photosensitizing properties and have been used as antimicrobial agents to photoinactivate microorganisms, including multiresistant *Staphylococcus aureus* (MRSA) strains. It is well recognized that the combination of different therapeutic agents might improve the biological outcome. In this present work, a novel *meso*-arylporphyrin and its Zn(II) complex functionalized with sulfonamide groups were synthesized and characterized and the antibacterial activity towards MRSA with and without the presence of the adjuvant KI was evaluated. For comparison, the studies were also extended to the corresponding sulfonated porphyrin TPP(SO_3_H)_4_. Photodynamic studies revealed that all porphyrin derivatives were effective in photoinactivating MRSA (>99.9% of reduction) at a concentration of 5.0 μM upon white light radiation with an irradiance of 25 mW cm^−2^ and a total light dose of 15 J cm^−2^. The combination of the porphyrin photosensitizers with the co-adjuvant KI during the photodynamic treatment proved to be very promising allowing a significant reduction in the treatment time and photosensitizer concentration by six times and at least five times, respectively. The combined effect observed for TPP(SO_2_NHEt)_4_ and ZnTPP(SO_2_NHEt)_4_ with KI seems to be due to the formation of reactive iodine radicals. In the photodynamic studies with TPP(SO_3_H)_4_ plus KI, the cooperative action was mainly due to the formation of free iodine (I_2_).

## 1. Introduction

Sulfonamides or sulfa drugs were the first effective synthetic chemical entities used systematically against a broad spectrum of bacteria [1,2]. They act as competitive inhibitors of dihydropteroate synthase, mimetizing *p*-aminobenzoic acid and therefore inhibiting the synthesis of tetrahydrofolic acid, which is essential for the formation of nucleic acids precursors in bacteria [3]. Since their discovery in 1935 [4,5], many sulfonamides have been developed, but their importance has declined due to an increase in microbial resistance. However, recent strategies have focused on the development of novel treatment options and alternative antimicrobial therapies. Among them, antimicrobial Photodynamic Treatment (aPDT) has been described as a promising alternative to conventional antibiotics as it leads to bacterial cell death without the development of microbial-resistant strains [6,7]. This technique requires the administration of a photosensitizer (PS), that, after being photoactivated with visible light, interacts with dioxygen (O_2_) producing reactive oxygen species (ROS) via two different mechanisms. Singlet oxygen (^1^O_2_), the main ROS responsible for microbial death, is produced by mechanism Type II, while hydroxyl radicals (^•^OH), superoxide anion radical (O_2_^•−^), and H_2_O_2_ are produced by mechanism Type I [8]. These ROS can act in the external cell wall of bacterial cells leading to their inactivation. This therapeutic modality is being recognized as an effective method for inactivating a broad spectrum of microorganisms, including multidrug-resistant Gram-(+) and Gram-(–) bacterial strains [9,10,11,12,13,14,15,16,17,18,19,20].

Porphyrins are the most common PS used in aPDT due to their high efficacy against a large spectrum of microorganisms [21,22,23]. Furthermore, porphyrins are easily modified allowing the insertion of groups to improve their photophysical properties and affinity to microbial cells. Research efforts have been focused on the development of cationic porphyrin derivatives that clearly demonstrates aPDT inactivation efficacy [24,25,26,27,28]. Their biocide effect arises from the possibility of the interaction of these compounds with the bacterial cell membrane [29,30,31,32,33,34]. In contrast, the development of anionic porphyrins, regardless of their efficiency to produce ROS, does not receive great attention because of their low interaction with the bacterial membrane [35,36,37]. However, negatively charged functional groups induce an inherent solubility of the porphyrins in aqueous media, which would greatly benefit their biological application [38,39]. In addition to negative and positive PS, neutral PS has also been studied and it was found that amphiphilicity can promote the enhancement of cell membrane interaction, playing an important role in the therapeutic effect [8]. The functionalization of organic compounds with sulfonamides is a versatile method to modulate amphiphilicity [5,40]. For instance, it was reported that the insertion of sulfonamide groups at the periphery of phthalocyanines can enhance their efficacy against bacteria [41], even suggesting that the presence of these groups may increase the interaction of the PS-sulfonamide conjugates towards target components of the bacterial cells’ outer structures. Porphyrins and metalloporphyrins containing sulfonamides and positively charged peripheral *N*-alkyl pyridinium groups have also been prepared and considered for biological applications such as DNA binding and intercalation [42,43]. Furthermore, synthetic strategies for preparing libraries of porphyrin-sulfonamides as photosensitizing agents have been disclosed in the literature [40,44,45,46,47]. However, as far as we know, their biological potential as aPDTagents is as yet poorly evaluated.

In 2017, Michael Hamblin and co-workers [48] unveiled the potentiation of aPDT(up to 6 logs of extra killing) by the addition of salts (KI, KBr, NaSCN, NaN_3_, NaNO_2_). The most powerful and versatile salt was potassium iodide (KI). This salt can enhance the photodynamic efficiency of neutral and cationic porphyrins resulting in a higher inactivation rate when compared to the use of PS alone. This synergic effect is caused by the initial production of peroxyiodide (HOOI_2_^–^) as a result of the interaction of KI with ^1^O_2_ produced by the PS. Then, the peroxyiodide can undergo further degradation into cytotoxic species like free iodine/tri-iodine ion (I_2_/I_3_^–^) and hydrogen peroxide (H_2_O_2_) or iodine radicals (I_2_^–•^) [18,22,49]. Several photodynamic studies were performed since then to improve antimicrobial outcomes [16,50,51,52,53]. In this context, Hamblin and co-workers selected the tetracationic 5,10,15,20-tetrakis(1-methylpyridinium-4-yl)porphyrin tetratosylate (**TMPyP**) and the tetra-anionic 5,10,15,20-tetrakis(4-sulfonatophenyl)porphyrin dihydrochloride [**TPP(SO_3_H)_4_.2HCl**] in order to obtain knowledge about porphyrin net charges and the effect of KI as a co-adjuvant in bacteria targeting [54]. **TPP(SO_3_H)_4_·2HCl** (200 nM) under blue light at 415 nm at an irradiance of 50 mW cm^−2^ and a total light dose of 10 J cm^−2^ in the presence of KI (100 mM) was more effective than **TMPyP** in eradicating the Gram-(+) bacterium, methicillin-resistant *Staphylococcus aureus* (MRSA) and the fungal yeast *Candida albicans*. Since **TPP(SO_3_H)_4_** was used in the dihydrochloride salt form, the authors explained the effectiveness of **TPP(SO_3_H)_4_.2HCl** by the existence of a degree of cationic character when in the presence of bacteria [54].

Considering the potential of porphyrins as PSs and sulfonamides as bacteriostatic compounds, we decided to take advantage of both chemical entities, by preparing porphyrins containing sulfonamides for bacterial photoinactivation. The capability of the porphyrin core to complex with a large number of metal cations, giving metalloporphyrins, strongly influences their electronic and photophysical properties by internal heavy-atom effects [55], enhancing the intersystem crossing to the triplet state which is the precursor to the formation of ROS and ^1^O_2_ [56]. Thus, the present work describes the synthetic access and photodynamic action of the free-base 5,10,15,20-tetrakis[4-(*N*-ethylsulfamoyl)phenyl]porphyrin (**TPP(SO_2_NHEt)_4)_** and of its zinc(II) complex, **ZnTPP(SO_2_NHEt)_4_** towards the methicillin-resistant *Staphylococcus aureus* (MRSA), selected as a bacterial model of Gram-(+). The biological studies performed in the presence of these PSs bearing four sulfonamide substituents at the *para* position of each phenyl ring were carried out with and without the adjuvant KI. For comparison, the 5,10,15,20-tetrakis(4-sulfophenyl)porphyrin, **TPP(SO_3_H)_4_** just bearing sulfonic acid groups was also considered. An investigation into the photophysical/photochemical features of the molecules under study was also carried out in order to corroborate their photodynamic action.

## 2. Results

### 2.1. Synthesis and Structural Characterization of Meso-Aryl Porphyrins Functionalized with Sulfonamides and Sulfonic Acids

The required porphyrins functionalized with sulfonamides and sulfonic acid substituents were obtained from 5,10,15,20-tetraphenylporphyrin (TPP) prepared according to the literature (Figure 1) [57,58]. To synthesize the required amphiphilic porphyrin sulfonamide derivative, the functionalization of **TPP** into the corresponding chlorosulfonated derivatives was achieved by mixing **TPP** with an excess of chlorosulfonic acid at room temperature over one hour [44,45,46]. After reaction completion and work-up, the **TPP(SO_2_Cl)_4_** derivative was obtained in a quantitative yield, without chromatographic purification.

The nucleophilic step was performed at room temperature for 3 h between the **TPP(SO_2_Cl)_4_** dissolved in dichloromethane and a large excess of ethylamine to afford the corresponding sulfonamide derivative. After work-up, and silica gel chromatography, the amphiphilic sulfonamide derivative 5,10,15,20-tetrakis[4-(*N*-ethylsulfamoyl)phenyl]porphyrin (**TPP(SO_2_NHEt)_4_**) was isolated with 88% yield. The compound structure was confirmed by ^1^H NMR (Appendix A) and high-resolution mass spectrometry (HRMS) (Appendix A). With the aim of modulating the photophysical and photochemical features, the complexation of the free base sulfonamide, **TPP(SO_2_NHEt)_4_** with zinc(II) acetate was carried out in chloroform/methanol [59,60]. After confirming the success of the metalation process, by UV-Vis and TLC, the organic layer was washed with distilled water, giving 5,10,15,20-tetrakis[4-(*N*-ethylsulfamoyl)phenyl]porphyrinatezinc(II) (**ZnTPP(SO_2_NHEt)_4_**) quantitatively. The structure of the complex was confirmed by UV-Vis analysis, ^1^H NMR (Appendix A), and HRMS (Appendix A).

The **TPP(SO_2_Cl)_4_** was also used to prepare the **TPP(SO_3_H)_4_** quantitatively, suspending the chlorosulfonated derivative in water under reflux for 12 h (see Figure 1) [61]. The ^1^H NMR confirmed the chemical structure of **TPP(SO_3_H)**_4_ which is in accordance with the literature (Appendix A) [44].

### 2.2. Photophysical Characterization

#### 2.2.1. UV-Vis Absorption Properties

The absorption spectra of all prepared compounds show the typical profile of *meso*-tetraarylporphyrins (Figure 1), which are dominated by a Soret band (B band) around 420 nm and the Q bands varying between four and two depending on the macrocycle symmetry (Table 1). The free-base derivatives **TPP(SO_3_H)_4_** and **TPP(SO_2_NHEt)_4_** present four Q bands with relative intensities of “etio” type while the complex **ZnTPP(SO_2_NHEt)_4_** displays the expected two Q bands between 560–600 nm (Table 1).

The molar absorption coefficients of the prepared PS were determined from the Beer–Lambert Law using DMF solutions and are presented in Table 1.

#### 2.2.2. Fluorescence Spectroscopy

The emissive features of the prepared molecules were evaluated and Figure 2 depicts the recorded emissive spectra of all derivatives including **TPP** in DMF as a reference. All spectra show two emission bands: a band at a shorter wavelength attributed to the Q (0,0) transition, and the adjacent band attributed to the Q (0,1) transition (Table 1). Looking at the relative intensity of the bands, it is shown that the intensity of the first band is always higher than the second one whose relative intensity increases when **TPP(SO_2_NHEt)**_4_ is complexed with zinc(II). The free-base derivatives **TPP(SO_3_H)_4_** and **TPP(SO_2_NHEt)**_4_ showed similar fluorescence quantum yields (Φ_F_ ~0.13) to **TPP** (Φ_F_ = 0.11). However, the zinc(II) complex **ZnTPP(SO_2_NHEt)_4_** displays a lower fluorescence quantum yield (Φ_F_ = 0.04).

#### 2.2.3. Singlet Oxygen Generation

The generation of ^1^O_2_ was assessed by using a well-established method [64]. The photo-oxidation of 9,10-dimethylanthracene (DMA) was followed by UV-Vis spectroscopy in the presence of each porphyrin derivative upon irradiation at 420 ± 5 nm, in DMF. The photodecay produced by the generation of ^1^O_2_ follows a pseudo-first-order kinetic and Figure 3 shows the results as ln(A_0_/A)*f(t).*

#### 2.2.4. Detection of Iodine Formation

The ability of **TPP(SO_2_NHEt)_4_**, **ZnTPP(SO_2_NHEt)_4_)_,_** and **TPP(SO_3_H)_4_** to generate iodine (I_2_) was analyzed, and the assays were performed by monitoring its absorbance at 340 nm, upon white light irradiation of each porphyrin derivative (25 mW cm^−2^) at a concentration of 5.0 µM in the presence of KI at a concentration of 100 mM (Appendix A). The results in Appendix A show that the highest amount of I_2_ was observed in the presence of **TPP(SO_3_H)_4_**, which attained the maximum after ca. 15 min of irradiation. Although in lower amounts, the free-base **TPP(SO_2_NHEt)_4_** was also able to generate I_2_, which slowly increased in the course of the experiment. The formation of I_2_ by the complex **ZnTPP(SO_2_NHEt)**_4_ was negligible.

### 2.3. Photodynamic Inactivation of MRSA

In this study, the photodynamic efficiency of **TPP(SO_3_H)_4_**, **TPP(SO_2_NHEt)_4_**, and **ZnTPP(SO_2_NHEt)_4_** was evaluated against MRSA, selected as a Gram-(+) bacterial model. We also evaluated the combined effect of each PS with KI, a well-known co-adjuvant of aPDT [48], allowing in some cases to reduce the treatment time and the concentration of PS applied.

Photodynamic assays were performed in the presence of the PSs alone at 1.0 µM and 5.0 µM [and also at 0.1 µM and 0.5 µM for **TPP(SO_3_H)_4_**] and in combination with KI at 100 mM, for 120 min of white light irradiation (400–700 nm) at an irradiance of 25 mW cm^−2^. The results obtained for the PSs **TPP(SO_3_H)_4_**, **TPP(SO_2_NHEt)_4_**, and **ZnTPP(SO_2_NHEt)_4_** are represented in Figure 4, Figure 5 and Figure 6, respectively.

The results obtained indicate that the porphyrinic PSs were efficient in reducing the viability of MRSA and that the inactivation profiles were PS- and concentration-dependent (Figure 4, Figure 5 and Figure 6). In the case of **TPP(SO_3_H)_4_** at 5.0 µM (Figure 4a), a decrease in MRSA viability was observed until the detection limit of the method (~7.5 Log reduction of CFU mL^−1^) after 10 min (light dose of 15 J cm^−2^) of photodynamic treatment (ANOVA, *p* < 0.0001). The same effect was achieved after 30 min of treatment (light dose of 45 J cm^−2^) at a concentration of 1.0 µM of **TPP(SO_3_H)_4_** (Figure 4b). The combination of **TPP(SO_3_H)_4_** at 5.0 and 1.0 µM with KI at 100 mM improved the photodynamic effect of this PS and in both cases allowed for a reduction in the treatment time to 5 min (light dose of 7.5 J cm^−2^) (minimal time point considered) (ANOVA, *p* < 0.0001). These results indicate that the **TPP(SO_3_H)_4_** + KI combination is promising and allows not only a reduction in the treatment time by about 6 times but also a reduction in the PS concentration to at least 1.0 µM. The outstanding efficiency of **TPP(SO_3_H)_4_**, towards MRSA strain at 5.0 and 1.0 µM and the short time required for reducing the viability of MRSA until the method detection limit, led us to investigate the inactivation profile at lower PS concentrations (0.1 µM and 0.5 µM). When a concentration of 0.5 µM of **TPP(SO_3_H)_4_** was used (Figure 4c), a decrease in MRSA viability was observed until the detection limit of the method (~7.5 Log reduction of CFU mL^−1^) after 60 min (light dose of 90 J cm^−2^) of photodynamic treatment (ANOVA, *p <* 0.0001). Moreover, when 0.1 µM of PS was employed, the reduction viability of MRSA was not so effective: ~5.1 Log reduction of CFU mL^−1^ (99.999%), after 120 min of irradiation (light dose of 180 J cm^−2^). Still, the combination of **TPP(SO_3_H)_4_** at 0.5 and 0.1 µM with KI at 100 mM (Figure 4c,d) improved the photodynamic effect of this PS, and in both cases allowed a reduction in the treatment time to 15 min and 30 min (light doses of 22.5 and 45 J cm^−2^), respectively, to reach the method detection limit (ANOVA, *p <* 0.0001).

**ZnTPP(SO_2_NHEt)_4_** still proved to be a promising compound for inactivating MRSA (Figure 5). At 5.0 and 1.0 µM, this PS led to a total reduction of bacterial viability until the method detection limit (~7.5 log CFU mL^−1^ reduction) after 45 min and 90 min of treatment (light doses of 67.5 and 135 J cm^−2^), respectively (ANOVA, *p <* 0.0001) (Figure 5a,b). In the presence of KI, the effect of PS was also improved, allowing the treatment time to be reduced to 15 and 45 min (light doses of 22.5 and 67.5 J cm^−2^), at a PS concentration of 5.0 and 1.0 µM, respectively (ANOVA, *p <* 0.0001). These results indicate that the use of KI as a co-adjuvant not only allowed a reduction in the treatment by about 3 times (from 45 to 15 min) but also reduces by five times the concentration of PS required to reach the method detection limit (from 5.0 to 1.0 µM).

The **TPP(SO_2_NHEt)_4_** showed a lower photodynamic activity rate (ANOVA, *p <* 0.0001) (Figure 6); at 5.0 µM, the reduction in bacterial viability until the detection limit of the method (~7.5 Log of CFU mL^−1^ of reduction, *p <* 0.0001) was attained after 120 min of treatment (light dose of 180 J cm^−2^) (Figure 6a). At 1.0 µM and under the same irradiation conditions a lower decrease in MRSA, by about ~3.4 Log CFU mL^−1^ (ANOVA, *p <* 0.0001), was observed (Figure 6b). The combination of **TPP(SO_2_NHEt)_4_** + KI resulted in faster bacterial inactivation rates until the detection limit of the method: at 5.0 µM the time required of white light irradiation was 30 min while at 1.0 µM was 60 min (light doses of 45 and 90 J cm^−2^) (Figure 6a,b) (ANOVA, *p <* 0.0001). It is noteworthy that the combination of **TPP(SO_2_NHEt)_4_** (1.0 µM) + KI was more effective in inactivating MRSA than the PS alone at 5.0 µM. These results indicate that KI was effective in improving the effect of **TPP(SO_2_NHEt)_4_**, allowing a decrease in the treatment time by about four-fold, and also the PS concentration by five-fold.

For all assays, light controls (LC), KI controls (LC+KI), and dark controls performed for the tested PS (DC) showed no significant variation in the number of viable MRSA cells (ANOVA, *p* > 0.05). These results indicated that the bacterial viability was not affected by light and other test conditions, that KI is not toxic at the concentration tested throughout the treatment time, and the PSs did not induce cytotoxicity in the dark at the highest concentration tested (5.0 µM) (Figure 4, Figure 5 and Figure 6).

## 3. Discussion

The synthetic route to obtain the porphyrin sulfonamides **TPP(SO_3_NHEt)_4_** and **ZnTPP(SO_3_NHEt)_4_** and the sulfonic acid derivative **TPP(SO_3_H)_4_** using 5,10,15,20-tetraphenylporphyrin (TPP) as starting material, was planned according to well-established procedures and comprise the following steps: (i) chlorosulfonation of **TPP** to the corresponding 5,10,15,20-tetrakis(*p*-chlorosulfophenyl)porphyrin derivative **TPP(SO_2_Cl)_4_**; (ii) reaction of this derivative with water or ethylamine to give the water-soluble sulfonic acid derivative **TPP(SO_3_H)_4_** or the amphiphilic sulfonamide porphyrin **TPP(SO_3_NHEt)_4_**, respectively, and (iii) complexation of the free base sulfonamide with the Zn(II) acetate to afford the metalloporphyrin complex **ZnTPP(SO_3_NHEt)_4_** (Figure 1).

The introduction of sulfonate groups in aromatic compounds [65] is a reaction widely used in the functionalization of organic molecules [38]. However, one of the hurdles with these reactions is that very polar sulfonic groups are obtained, the purification of which becomes quite difficult. Therefore, the chlorosulfonation reaction is an alternative for the peripheral functionalization of porphyrins, as the chlorosulfonates formed are stable enough to further react in the presence of many nucleophiles such as water [44,61], amines [46], or alcohols [45,66] affording porphyrins with sulfonic acids, sulfonamides or sulfo esters moieties with high biological interest [41,45,46,67,68]. The TPP chlorosulfonation reaction was carried out and we proceed with our synthetic strategy of functionalization of porphyrins with the aim of obtaining PS with different physicochemical properties, such as different solubilities in physiological or aqueous media. Due to the high reactivity of chlorosulfonates with amines, and as we wanted to develop amphiphilic PS, this led us to prepare porphyrins functionalized with sulfonamide groups. We added an excess of ethylamine to **TPP(SO_2_Cl)_4_** dissolved in CH_2_Cl_2_, yielding **TPP(SO_2_NHEt)_4_**. To modulate the photophysical and photochemical features, it was decided to prepare porphyrin metal complexes since it is well established that the coordination of metals in tetrapyrrolic macrocycles leads to an increase in the production of ^1^O_2_ and other ROS, owing to the heavy atom effect that potentiates formation of the triplet state (^3^PS*) and reduces the quantum yield of fluorescence (Φ_F_) [69,70]. Following this, a complexation reaction of the sulfonamide **TPP(SO_2_NHEt)_4_** with zinc(II) salts was carried out, yielding purple crystals of **ZnTPP(SO_2_NHEt)_4_**. When **TPP(SO_2_Cl)_4_** was hydrolyzed with water, under reflux, the water soluble **TPP(SO_3_H)_4_**. was obtained quantitatively.

The UV-Vis absorption (Figure 1) as well as the fluorescence emission (Figure 2) features and the fluorescence quantum yield of the studied PS are presented in Table 1. It is shown that the sulfonic acids or sulfonamide groups at the periphery of **TPP** do not cause a significant change in the UV-Vis spectra when compared with the starting *meso*-phenylporphyrin, **TPP**. Moreover, the complexation of **TPP(SO_2_NHEt)_4_** with zinc(II) acetate, **ZnTPP(SO_2_NHEt)_4_** changes the free base symmetry from D_2h_ to D_4h_, and, consequently, the four Q bands are converted in two for the metal complex [69]. Additionally, complexation with zinc(II) leads to a redshift (~7 nm) of the Soret band. This shift is typical due to an interaction of the metal ion *d* orbital with the porphyrin aromatic π system [69].

All fluorescence emission spectra of *meso*-substitutedporphyrins display similar profiles. The fluorescence quantum yields calculated for the free base porphyrins **TPP(SO_3_H)_4_** and **TPP(SO_2_NHEt)_4_** are very similar to the values found for **TPP**, which indicates that the introduction of sulfonic acids or sulfonamides onto the peripheral phenyl rings does not affect significantly the singlet state properties (^1^S*). Conversely, the presence of zinc(II) led to a significant decrease in the fluorescence quantum yield compared to **TPP** and **TPP(SO_2_NHEt)_4_**. It is well-established from the literature [69,71] that metalloporphyrins with closed metal shells, such as Zn(II), are less fluorescent than the corresponding free-base porphyrins, have shorter fluorescence lifetimes, and higher intersystem crossing to triplet states, promoted by the spin-orbit coupling mechanism (heavy atom effect) [55].

The success of the prepared porphyrin derivatives as PS depends strongly on their capability to generate ROS, such as ^1^O_2_, the main oxidative species accountable to cause microorganisms inactivation [8,72]. The ^1^O_2_ formation of the studied PS was determined by the photo-oxidation of 9,10-dimethylanthracene (DMA) [64]. ^1^O_2_ origins the photo-oxidation of DMA, and its conversion to endoperoxide. In this sense, it is possible to track the decrease in DMA absorption in solution at 378 nm. As observed in Figure 3, all the evaluated PSs were able to produce ^1^O_2_ and no photo-oxidation of DMA was observed in their absence. In this series, **ZnTPP(SO_2_NHEt)_4_** merits to be highlighted since is the most efficient compound to generate ^1^O_2_. This observation is in accordance with the literature data, since it is expected that zinc(II) complexes are better ^1^O_2_ producers due to the heavy atom effect, as stated above [69,70]. **TPP(SO_3_H)_4_** and **TPP(SO_2_NHEt)_4_** revealed similar ^1^O_2_ generation as the reference TPP which is considered a good ^1^O_2_ generator (Φ_Δ_ = 0.65 in DMF) [63]. Therefore, all molecules are considered promising for application as PS in aPDT.

aPDT relies upon the use of a PS molecule whose structure and substituent groups are highly important for the antimicrobial success of this approach [18,56,73,74]. A few literature studies reported the use of sulfonamide phthalocyanine [41] and *ortho*-halogenated sulfonamide porphyrin (containing a sulfonamide at the 3′ position of the phenyl ring) [68,75] as PS against bacteria. However, the aPDT effect on bacteria using 4′-sulfonamide groups on non-halogenated porphyrins is yet to be studied.

Co-adjuvant KI has been reported to potentiate the photodynamic efficiency of a broad spectrum of molecules [22,76]. The mechanism under the potentiation of photodynamic efficiency is related to the I_2_ formation since ^1^O_2_ reacts with KI producing peroxyiodide (HOOI_2_^–^). This species can be further decomposed into free iodine (I_2_/I_3_^–^) and iodine radicals (I_2_^•–^), which are responsible to cause cell killing [22,48,76]. Thus, the formation of iodine by the prepared PS at 5.0 µM was evaluated in the presence of KI at 100 mM, under irradiation with white light (Appendix A). In the presence of KI the absorbance at 340 nm of all solutions increased as a result of I_2_ formation, although **TPP(SO_3_H)_4_** remarkably stands out with a significative absorbance increase in the initial 15 min of irradiation being the most efficient PS to generate I_2_, followed by **TPP(SO_2_NHEt)_4_** and the less efficient compound **ZnTPP(SO_2_NHEt)_4_** to generate I_2_. Considering all the photophysical data, the *meso*-arylporphyrins functionalized with sulfonamide groups **TPP(SO_2_NHEt)_4_** and **ZnTPP(SO_2_NHEt)_4_** and the sulfonic acid derivative **TPP(SO_3_H)_4_** proceeded to biological evaluation due to it promising properties as PS. Thus, all three derivatives were assessed for their photodynamic action in combination with the adjuvant KI against a methicillin-resistant *Staphylococcus aureus* (MRSA) strain, selected as a bacterial model of Gram-(+) bacteria.

Our findings indicate that, at low concentrations, the **TPP(SO_3_H)_4_** (5.0, 1.0, 0.5, and 0.1 µM), **TPP(SO_2_NHEt)_4_** (5.0 and 1.0 µM), and **ZnTPP(SO_2_NHEt)_4_** (5.0 and 1.0 µM) were effective in the photoinactivation of MRSA strain, promoting a bacterial reduction always higher than 3.0 Log CFU mL^−1^ (> 99.9%, ANOVA, *p <* 0.0001) (Figure 4, Figure 5 and Figure 6). Thus, according to the guideline of the American Society for Microbiology, the PSs synthesized in this study meet the necessary conditions to be considered bactericidal agents [77]. Comparing the photodynamic efficiency of all prepared PS, **TPP(SO_3_H)_4_** proved to be the most effective in reducing the viability of MRSA cells, followed by **ZnTPP(SO_2_NHEt)_4_** and finally by **TPP(SO_2_NHEt)_4_** (ANOVA, *p <* 0.0001). It was clear that the **ZnTPP(SO_2_NHEt)_4_** was more efficient for the inactivation of MRSA when compared with the parent free-base **TPP(SO_2_NHEt)_4_**. The different effectiveness rates exhibited by the tested PSs might be attributed to their physicochemical properties. It is well known that the aggregation behavior of a PS when in physiological media tends to significantly diminish the amount of ^1^O_2_ produced, which leads to a reduction in the photoinactivation rate [78]. The **TPP(SO_3_H)_4_** is the most soluble PS in an aqueous solution and consequently less affected by aggregation phenomena, which can explain its greater effectiveness in the photoinactivation of MRSA cells. The two sulfonamide-porphyrin derivatives **TPP(SO_2_NHEt)_4_** and **ZnTPP(SO_2_NHEt)_4_** are amphiphilic compounds and when in aqueous solution (PBS) some aggregation phenomena might occur, which substantiates the decrease in inactivation rate when compared with **TPP(SO_3_H)_4_**. This phenomenon can explain why **ZnTPP(SO_2_NHEt)_4_** with the highest efficiency to generate ^1^O_2_ presented a slower rate of inactivation towards the MRSA strain than **TPP(SO_3_H)_4_** (Figure 3). Even so, the photodynamic effect of all synthesized PS is even more remarkable when compared with those of the literature [68,75]. In 2017, Dabrowski [68] and co-authors studied the photodynamic efficacy of fluoro and chloro, *ortho*-halogenated porphyrins containing sulfonic acid groups and sulfonamides at the 3′ position: 5,10,15,20-tetrakis(2,6-difluoro-3-sulfophenyl)porphyrin (**F_2_POH**), 5,10,15,20-tetrakis(2,6-dichloro-3-sulfophenyl)porphyrin (**Cl_2_POH**), and 5,10,15,20-tetrakis[2,6-dichloro-3-(*N*-ethylsulfamoyl)phenyl]porphyrin (**Cl_2_PEt**). These PSs were studied against Gram-(+) (*S. aureus*, *E. faecalis*), Gram-(−) bacteria (*E. coli*, *P. aeruginosa*, *S. marcescens*), and fungal yeast (*C. albicans*). The best photoinactivation results against *S. aureus* were found for both sulfonic acid derivatives (**F_2_POH**, **Cl_2_POH)** at a concentration of 20 μM and after 10 min irradiation (10 J cm^−2^) reaching a 6 Log CFU mL^−1^ reduction. In addition, for the same strain and at a concentration of 20 μM the *ortho*-halogenated porphyrins, **Cl_2_PEt** gave only a reduction of 99% (2 Log CFU mL^−1^ reduction) in the *S. aureus* survival after 10 min irradiation. Later on, the same group [75] reported a comparison study between neutral (**TPP**), positive (**TMPyP**), and negative (**TPPSO_3_H**) non-halogenated *meso*-arylporphyrins, with a series of neutral (**ClTPP**, **Cl_2_TPP**) or anionic (**Cl_2_TPPS**) *ortho*-chlorophenylporphyrins. The photodynamic treatments with the **Cl_2_TPPS** used as PS at a concentration of 20 μM led to the 5 Log of CFU mL^−1^ reduction of *S. aureus* after to have received a low light dose (5 J cm^−2^). Moreover, high efficiency in the *S. aureus* photoinactivation was found when 100 mM of KI was used as a co-adjuvant, and the photodynamic inactivation was enhanced by 2–3 Log for *S. aureus*. When the authors increased the light dose to 40 J cm^−2^, a complete destruction of Gram-(+) bacterium was observed.

In the present study, the porphyrinic PSs evaluated were more efficient against *S. aureus* since the bacterial viability reached the detection limit of the methodology (reduction of 7.5 Log of CFU mL^−1^) with a lower concentration of PS (5.0 µM without co-adjuvant KI) and a total light dose ranging from 15 to 180 J cm^−2^ [**TPP(SO_3_H)_4_:** 15 J cm^−2^; **ZnTPP(SO_2_NHEt)_4_**: 67.5 J cm^−2^; **TPP(SO_2_NHEt)_4_**: 180 J cm^−2^].

Although the synthesized PSs were effective in inactivating MRSA, their photodynamic effect was significantly improved in the presence of the co-adjuvant KI. Previous studies have reported that the bacterial death curve can give an indication of the iodine species responsible for the extra inactivation [22,79]. When the death curve assumes an abrupt bacterial inactivation profile, free I_2_ is the main specie responsible for microbial inactivation. If there is a gradual increase in the bacterial death rate, there is a more pronounced contribution from short-lived reactive iodine species. In this way, the PS + KI inactivation profiles were evaluated to identify the potential species responsible for the additional bacterial killing. In the case of **TPP(SO_3_H)_4_** for all tested concentrations (0.1—5.0 µM) + KI, an abrupt decrease in the MRSA inactivation profile (reaching the detection limit of the method) was observed (5 min at 1.0 µM—30 min at 0.1 µM, Figure 4). These results can be explained by the preferential decomposition of peroxyiodide into free iodine (I_2_), a species recognized for its high antimicrobial action after reaching a threshold concentration, justifying the rapid inactivation of MRSA and the high efficacy of this combination. This production of free iodine (I_2_) was confirmed by the high and rapid detection of I_2_ by spectrophotometry (Appendix A).

In the case of the **TPP(SO_2_NHEt)_4_** + KI and **ZnTPP(SO_2_NHEt)_4_** + KI combinations, a gradual decrease in the MRSA inactivation profile was observed, mainly evident at the lowest concentration of PS (1.0 µM) (Figure 5 and Figure 6, respectively). These results lead us to assume that the mechanism of action is related to the preferential decomposition of peroxyiodide into iodine radicals which, due to their short diffusion distance, cause a gradual decrease in the inactivation profile. This fact was confirmed by the low detection of free I_2_ by spectrophotometric analysis, which did not affect the high combined effect observed with **TPP(SO_2_NHEt)_4_** and **ZnTPP(SO_2_NHEt)_4_** in the presence of KI.

## 4. Materials and Methods

The most relevant spectroscopic and structural features of the prepared porphyrin precursors are described in detail in the Appendix A. To the best of our knowledge, compounds **TPP(SO_2_NHEt)_4_ and ZnTPP(SO_2_NHEt)_4_** were synthesized here for the first time and the synthesis and full structural characterization of these compounds are described below.

### 4.1. Photosensitizers Preparation and Characterization

**5,10,15,20-tetrakis(4-chlorosulfonylphenyl)porphyrin, TPP(SO_2_Cl)_4_** was obtained following a literature method [46] through the aromatic electrophilic chlorosulfonation of **TPP**. This porphyrin was used directly (without further purification) in the subsequent reactions.

**5,10,15,20-tetrakis(4-sulfophenyl)porphyrin, TPP(SO_3_H)**_4_ was obtained by hydrolysis of TPP(SO_2_Cl)_4_ in water under reflux over 18 h [44] and the ^1^H NMR is in agreement with the literature. ^1^H NMR (300 MHz, DMSO-d_6_), δ ppm: 8.86 (s, 8H, β-H); 8.19 (d, *J* = 8.1 Hz, 8H, Ar-H); 8.05 (d, *J* = 8.1 Hz, 8H, Ar-H); -2.98 (s, 2H, NH).

**5,10,15,20-tetrakis[4-(*N*-ethylsulfamoyl)phenyl]porphyrin, TPP(SO_2_NHEt)_4_**. To a round bottom flask with **TPP(SO_2_Cl)_4_** (100.9 mg, 0.1 mmol) dissolved in dichloromethane (50 mL) an excess of ethylamine (1.2 mmol, 0.1 mL) was added. The reaction mixture was stirred for 3 h at room temperature. The reaction was monitored by TLC using dichloromethane/ethyl acetate (7:3) as the eluent. Then, the organic layer was washed three times with 1M HCl solution, followed by a neutralization procedure with a saturated solution of sodium hydrogen carbonate and dried over anhydrous sodium sulfate. After solvent evaporation under reduced pressure, the solid residue was purified on a silica gel column chromatography using dichloromethane/ethyl acetate (7:3) as the eluent. The main compound **TPP(SO_2_NHEt)_4_** (92.6 mg) was isolated, yielding 88%. ^1^H NMR (300 MHz, DMSO-d_6_), δ ppm: 8.87 (s, 8H, β-H); 8.47 (d, 8H, *J* = 8.3 Hz, Ar-H); 8.25 (d, 8H, *J* = 8.3 Hz, Ar-H); 7.93 (t, 4H, *J* = 5.8 Hz, -NH-); 3.08–3.17 (m, 8H, -CH_2_-), 1.16–1.24 (m, 12H, -CH_3_), -2.95 (s, 2H, NH). HRMS-ESI(+): m/z calcd for C_52_H_51_N_8_O_8_S_4_: 1043.2690 [M+H]^+^; found 1043.2707.

**5,10,15,20-tetrakis(4-(*N*-ethylsulfamoyl)phenyl)porphyrinate zinc(II), ZnTPP(SO_2_NHEt)_4_**. In a round bottom flask, 20 mg of **TPP(SO_2_NHEt)_4_** (0.019 mmol) were dissolved in chloroform (10 mL). The reaction mixture was heated to 50 °C and then 35 mg of zinc(II) acetate (0.19 mmol), previously dissolved in methanol (5 mL), was added. The reaction remained under heating (50 °C) and magnetic stirring, being controlled by UV-Vis. In the end, the reaction was cooled to room temperature, the solvent was removed under a vacuum and the solid was dissolved in dichloromethane. The organic phase was washed with distilled water (three times) and dried over anhydrous sodium sulfate. After evaporation, the purple solid formed was dried under a vacuum. The compound **ZnTPP(SO_2_NHEt)_4_** (21 mg, 0.019 mmol) was obtained in quantitative yield. ^1^H NMR (300 MHz, CDCl_3_/CD_3_OD), δ ppm: 8.83 (s, 8H, β -H); 8.38 (d, 8H, *J* = 8.4 Hz, Ar-H); 8.25 (d, 8H, *J* = 8.4 Hz, Ar-H); 3.30 (q, 8H, *J* = 7.3 Hz, -CH_2_-), 1.32 (t, 12H, *J* = 7.3 Hz, -CH_3_). ^13^C (75 MHz, DMSO-d_6_), δ ppm: 149.6, 147.6, 139.4, 134.9, 131.7, 125.0, 119.1, 40.2, 38.3, 15.0. HRMS-ESI(+): m/z calcd for C_52_H_49_N_8_O_8_S_4_Zn: 1105.1842 [M+H]^+^; found 1105.1817.

### 4.2. UV-Vis Absorption Spectroscopy

The UV-Vis spectra were acquired using a Shimadzu UV-2501PC spectrophotometer on 1 × 1 cm quartz optical cells in DMF as a solvent. The molar absorption coefficients were calculated through the Beer–Lambert law.

### 4.3. Fluorescence Quantum Yield Measurements

The fluorescence excitation and emission spectra were acquired at room temperature using a Horiba Spex Fluoromax 4 Plus Spectrofluorimeter with DMF as a solvent. The quantum fluorescence yield (Φ_F_) of all PSs was determined using **TPP** as a reference in DMF (Φ_F_ = 0.11) [63]. Solutions of **TPP** in DMF and each PS were freshly prepared, and the emission spectra were measured upon excitation at 420 nm of solutions with an OD of 0.02 in quartz cells with four faces of 1 x 1 cm of optical path.

### 4.4. Singlet Oxygen Generation

Solutions of each compound (**TPP(SO_3_H)_4_**, **TPP(SO_2_NHEt)_4_**, and **ZnTPP(SO_2_NHEt)_4_**) and **TPP** in DMF (2.5 mL) were placed in a quartz cuvette and the absorbance adjust to ≈ 0.1 at 420 nm. Then, 9,10-dimethylanthracene (DMA) in DMF was added at a concentration of 30 µM. The solutions were irradiated at 420 ± 5 nm and the absorbance of each solution was monitored at 378 nm, every 60 s across 600 s, in a UV-2501PC SHIMADZU spectrophotometer and registered in a first-order kinetic plot.

### 4.5. Detection of Iodine Formation

In a 96-well microplate were prepared solutions of each PS (**TPP(SO_3_H)_4_**, **TPP(SO_2_NHEt)_4_**, and **ZnTPP(SO_2_NHEt)_4_**) at a concentration of 5.0 µM in PBS into which were added a solution of KI at 100 mM in PBS. All solutions were incubated under stirring in the dark for 15 min and then irradiated with white light (400–700 nm) at an irradiance of 25 mW cm^−2^. The formation of iodine (I_2_) during the experiment progression was monitored by measuring the absorbance at 340 nm at different pre-defined irradiation times in a Synergy™ HTX Multi-Mode Microplate Reader from BioTek Instruments.

### 4.6. Light Source

An LED system (LUMECO, 30 W, 2000 lm) was used as an artificial white light (400–700 nm) source in the iodine (I_2_) formation assay as well as the photodynamic treatment assays. Before each assay, the LED system was placed above the samples at a distance that allows homogeneous irradiation of the samples at an irradiance of 25 mW cm^−2^. The irradiance was measured and adjusted to 25 mW cm^−2^ with a FieldMaxII–TOP energy meter combined with a PowerSens PS19Q (Coherent).

### 4.7. Biological Assays

#### 4.7.1. Photosensitizer and Potassium Iodide Stock Solutions

The stock solutions of **TPP(SO_3_H)_4_**, **TPP(SO_2_NHEt)_4_**, and **ZnTPP(SO_2_NHEt)_4_** were prepared at 500 µM in DMSO and kept in the dark. Prior to each assay, the PS stock solution was sonicated for 30 min at room temperature (ultrasonic bath, Nahita 0.6 L, 40 kHz).

Potassium iodide (KI) was purchased from Sigma-Aldrich (St. Louis, MO, USA) and KI solutions were prepared at 5 M in sterile PBS immediately before each assay.

#### 4.7.2. Photoinactivation Conditions

For the biological assays, the methicillin-resistant *S. aureus* strain DSM 25693 was selected as a Gram-(+) bacterial model. This strain produces the staphylococcal enterotoxins A, C, H, G, and I. The bacterium was maintained in the laboratory on Tryptic Soy Agar medium (TSA, Merck) at 4 °C. Before each assay, three colonies were transferred to 30 mL of Tryptic Soy Broth medium (TSB, Merck) and then incubated at 37 °C for 18 h under stirring (120 rpm). Then, 300 µL of the previously grown bacterial suspension was transferred to 30 mL of new TSB medium and incubated under the conditions described above, to promote the bacterial growth until the stationary phase (approximately 10^9^ CFU mL^−1^).

#### 4.7.3. Photodynamic Treatment Experiments

To evaluate the photodynamic inactivation efficiency of the synthesized PSs against MRSA, the PSs **TPP(SO_3_H)_4_**, **TPP(SO_2_NHEt)_4_**, and **ZnTPP(SO_2_NHEt)_4_** were tested at the concentrations of 5.0 and 1.0 μM, in combination with KI at 100 mM. The **TPP(SO_3_H)_4_** was also tested at concentrations of 0.5 and 0.1 μM. Photodynamic assays were performed for 120 min under irradiation with white light (400–700 nm) at an irradiance of 25 mW cm^−2^.

The bacterial culture (approximately 10^9^ CFU mL^−1^) was 100-fold diluted in phosphate-buffered saline (PBS) and distributed in 12-well plates. An appropriate volume of each PS and PS + KI was added to obtain the desired PS concentrations. The total volume used per well was 4 mL. Simultaneously, the following controls were also performed: light control (LC), containing only the bacterial suspension and PBS, exposed to light; light control with KI (LC KI), containing the bacterial suspension in PBS and KI, exposed to light; dark control (DC) containing the bacterial suspension in PBS, KI and PS at the maximum tested PS concentration (5.0 μM) and protected from light with aluminium foil during the experiment time course.

The samples and controls were incubated in the dark for 15 min, under stirring (120 rpm) and at room temperature, to promote PS binding to bacterial cells. Afterward, samples and light controls were irradiated for 120 min under agitation, and simultaneously, dark controls were protected from light. To evaluate the effect of the treatments on the MSRA cells, aliquots (150 µL) of each sample and control were collected at pre-defined periods: 0 (after the dark incubation period) and 5, 10, 15, 30, 45, 60, 90, and 120 min of treatment. The collected aliquots were then serially diluted in PBS, drop-plated (10 µL) in triplicate on TSA, and incubated at 37 °C for 18–24 h. Later, colonies were counted at the most appropriate dilution and expressed as Colony Forming Units per mL (CFU mL^−1^). For each condition tested, three independent assays, each assay in triplicate, were performed.

#### 4.7.4. Statistical Analysis

Statistical analysis was performed using the GraphPad Prism 9 program. Normal distributions were analyzed using the Kolmogorov–Smirnov test and the homogeneity of variance was verified using the Brown–Forsythe test. Differences between the results were evaluated by 2-way ANOVA and by Tukey’s multiple comparison tests. *p* values < 0.05 were considered significant. For each condition tested at least three independent assays were performed and each assay was in triplicate.

## 5. Conclusions

The sulfonamide derivatives **TPP(SO_2_NHEt)_4_** and **ZnTPP(SO_2_NHEt)_4_** were efficiently prepared by well-established synthetic procedures. Photodynamic studies revealed that **TPP(SO_3_H)_4_**, **TPP(SO_2_NHEt)_4_**, and **ZnTPP(SO_2_NHEt)_4_** were effective PSs in the photoinactivation of Gram-(+) MRSA. **TPP(SO_3_H)_4_** showed greater efficacy than the other tested PSs, followed by **ZnTPP(SO_2_NHEt)_4_** and **TPP(SO_2_NHEt)_4_**. These results seem to be related to the physicochemical properties of these compounds, which modulate the solubility in aqueous media, the ^1^O_2_ production, and the production of iodine reactive species when combined with KI. Furthermore, the potential of these compounds is evidenced by the low PS concentrations required to photoinactivate a multiresistant *S. aureus* strain and by the low irradiance and light dose needed to achieve total inactivation of MRSA, being an advantage for clinical and environmental applications of these PSs. The co-adjuvant of KI improved the photodynamic effect of the three PSs, allowing a reduction in the treatment time by about 6 times but also reducing the PS concentration to at least 1.0 µM. The combined effect observed for the **TPP(SO_3_H)_4_** + KI is mainly due to the formation of free iodine, but for **TPP(SO_2_NHEt)_4_** and **ZnTPP(SO_2_NHEt)_4_** the effect of its combination with the KI co-adjuvant is mainly due to the formation of reactive iodine radicals.

## Data Availability

Not applicable.

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
