# Peer review of "Sulfonamide Porphyrins as Potent Photosensitizers against Multidrug-Resistant Staphylococcus aureus (MRSA): The Role of Co-Adjuvants"

_molecules, 2023, doi:10.3390/molecules28052067_

Round 1
Reviewer 1 Report
Dear Editor,
my comments are below;
- for the compound TPP(SO2NHEt)4 , in the H-NMR spectra, ıt seems the aliphatic region is abit complicated, please define the each peak, and please re-pruficaiton needs to be done
-the same applies to the compound ZnTPP(SO2NHEt)4
- why is the zinc(II) complex ZnTPP(SO2NHEt)4 display the lower fluorescence quantum yield? the author should explain it
-excatly which light irridation was used? should explain it clearly
Author Response
- For the compound TPP(SO2NHEt)4, in the H-NMR spectra, it seems the aliphatic region is a bit complicated, please define each peak, and please re-purification needs to be done.
Answer: The compound was purified by column chromatography and then recrystallization was attempted to improve the purity. However, due to the lower solubility in several solvents, we have some difficulty in obtaining crystals. However, the product obtained was dried in a vacuum oven for several days and showed a single spot in the TLC control. So, we believe that the principal reason why the 1H NMR spectra are not so clear in the aliphatic region TPP(SO2NHEt)4 is due to the residual signal of deuterated solvent at 2.50 ppm (deuterated-DMSO) and the residual signal of water at 3.33 ppm accompanied by all the side bands associated with those large signals. In order to clarify the extra signals due to the NMR solvent they were adequately assigned.
- The same applies to the compound ZnTPP(SO2NHEt)4
Answer: As before, the solid obtained after purification of ZnTPP(SO2NHEt)4 was dried in a vacuum oven for several days and showed a single spot in the TLC control. Since ZnTPP(SO2NHEt)4 presented a fair solubility in typical NMR solvents, in this case, we were able to use deuterated chloroform with a small percentage of deuterated methanol. Thus, the two main peaks are attributed to deuterated chloroform and deuterated methanol. Other small solvent peaks might be attributed to some residual water, CD3OH or due to the coordination of Zn(II) with these solvents affording peaks at different chemical shifts. In order to clarify this aspect all the solvent signals have been identified in the NMR spectra and the labile ones with an *.
- Why is the zinc(II) complex ZnTPP(SO2NHEt)4 display the lower fluorescence quantum yield? the author should explain it.
Answer: We thank the reviewer to comment on this photophysical aspect. It is well known that metalloporphyrins with closed metal shells, such as Zn(II) are less fluorescent than the corresponding free-base porphyrin, present shorter fluorescence lifetimes, and higher intersystem crossing to triplet states promoted by the spin–orbit coupling mechanism. In order to clarify this aspect to the readers the following text and adequate references were added:
It is well-established form the literature [65,67] that metalloporphyrins with closed metal shells, such as Zn(II) are less fluorescent than the corresponding free-base porphyrin, have shorter fluorescence lifetimes, and higher intersystem crossing to triplet states promoted by the spin–orbit coupling mechanism (heavy atom effect) [68].
- Exactly which light irradiation was used? should explain it clearly
Answer: The light irradiance used was 25 mW cm-2 and it was mentioned in the manuscript. Also, the light source used in photodynamic treatment assays and in the iodine formation was described in section 4.6.
4.6. Light source
A LED system (LUMECO, 30 W, 2000 lm) was used as artificial white light (400-700 nm) source in the iodine (I2) formation assay as well as at photodynamic treatment assays. Before each assay, the LED system was placed above the samples at a distance that allows homogeneous irradiation of the samples at an irradiance of 25 mW cm-2. The irradiance was measured and adjusted to 25 mW cm-2 with an energy meter FieldMaxII–TOP combined with a PowerSens PS19Q (Coherent).
Nevertheless, we revised the manuscript in order to be sure that the light irradiance used in each assay was mentioned.
Reviewer 2 Report
The paper is well organized and well written, with figures that clearly illustrate the results obtained and the conclusions describe as well as the results obtained.
This work presents the synthesis of a series of porphyrin dyes as well as their phototherapeutic potential against bacteria.
I leave below a few notes that the authors should take into account in order to improve this paper.
It is not clear which are the new compounds and which have already been synthesized and published. All new compounds should be characterized by means of 1H and 13C as well as HR-MS.
Singlet Oxygen generation was assessed by using DMA method. The authors should report the results obtained with the same method of a well known Singlet Oxygen generator probe for a comparison: methylene blue, Bengal rose etc.
The light treatment on the bacteria is very long, did the authors check if there is an increase of the temperature due to this long treatment?
Figure 4. Photodynamic inactivation of MRSA in the presence of TPP(SO3H)4. Which is the difference between the points in blue and light blue? I suppose the presence of KI, please correct the legend.
Author Response
Reviewer 2
The paper is well organized and well written, with figures that clearly illustrate the results obtained and the conclusions describe as well as the results obtained.
This work presents the synthesis of a series of porphyrin dyes as well as their phototherapeutic potential against bacteria.
I leave below a few notes that the authors should take into account in order to improve this paper.
Comment: The authors thank the reviewer for the recognition of the work value and the way it was presented.
- It is not clear which are the new compounds and which have already been synthesized and published. All new compounds should be characterized by means of 1H and 13C as well as HR-MS.
Answer: The new compounds are TPP(SO2NHEt)4 and ZnTPP(SO2NHEt)4. TPP(SO3H)4 was synthesized by us but it is not new. A sentence was included in the manuscript mentioning the new compounds. TPP(SO2NHEt)4 and ZnTPP(SO2NHEt)4 were characterized by 1H NMR and HRMS as was shown in the manuscript and in Supplementary Material. ZnTPP(SO2NHEt)4 was also characterized by 13C NMR. However, the solubility of TPP(SO2NHEt)4 was very low and it was not possible to obtain a 13C NMR spectra with enough intensity to differentiate the carbon signals from the spectrum baseline, even after several attempts and with acquisition experiments of ~48 h. Moreover, since for compound ZnTPP(SO2NHEt)4 the 13C NMR was obtained with acceptable definition and the ZnTPP(SO2NHEt)4 was prepared from TPP(SO2NHEt)4 (complexation with Zinc(II) Acetate) if the 13C NMR of this metallated compound is correct, we must conclude that the one of TPP(SO2NHEt)4 free-base is also correct.
2-Singlet Oxygen generation was assessed by using DMA method. The authors should report the results obtained with the same method of a well-known Singlet Oxygen generator probe for a comparison: methylene blue, Bengal rose etc.
Answer: We would like to point out that in these studies it was used as a reference the 5,10,15,20-tetraphenylporphyrin (TPP, also known as H2TPP) a well-known singlet oxygen generator (FD = 0.65 in DMF). This aspect was mentioned in the discussion section of the manuscript (page 10, lines 340-342).
TPP(SO3H)4 and TPP(SO2NHEt)4 revealed similar 1O2 generation as the reference TPP which is considered a good 1O2 generator (FD = 0.65 in DMF) [63]. Thus, all molecules are considered promising for application as PS in aPDT.
We would like to refer that the DMA method is used in our group for over a decade and is a very reliable method. Moreover, the method was validated by other groups for the determination of porphyrinic PS. See as examples:
https://link.springer.com/article/10.1007/s10895-020-02513-2
https://pubs.acs.org/doi/full/10.1021/acs.biomac.8b00605
3-The light treatment on the bacteria is very long, did the authors check if there is an increase in the temperature due to this long treatment?
Answer: We would like to refer that the temperature did not increase significantly during the aPDT treatment. Nevertheless, it is worth mentioning that the light control used in each assay (incubated in the same conditions as the sample) allows us to verify the effect of light and potential thermal effects on the bacteria viability. As we can verify no change the bacterial viability was found during the aPDT experiments after light irradiation of bacterial suspensions without PS (LC-light control) (Figures 4, 5 and 6).
4- Figure 4. Photodynamic inactivation of MRSA in the presence of TPP(SO3H)4. Which is the difference between the points in blue and light blue? I suppose the presence of KI, please correct the legend.
Answer: Thank you for noticing this typo. Figure 4 was correct: Blue corresponds to TPP(SO3H)4+ KI and light blue corresponds to TPP(SO3H)4.
Reviewer 3 Report
The manuscript "Sulphonamide Porphyrins as Potent Photosensitizers against Multidrug-Resistant Staphylococcus aureus (MRSA): the Role of co-Adjuvants" submitted by Sarabando er al. is an interesting contribution to the field of photodynamic therapy/photodynamic inactivation. The authors present two new compounds, a free base and the corresponding zinc complex of a tetrakis-sulfonamide-substituted porphyrin. They rely heavily on literature procedures for the precursor materials. I think the manuscript is suitable for publication in Molecules after some corrections or clarifications:
1. In Fig. 1b, a shoulder appears at the Qy(0,0) transition of the free base sulfonamide porphyrin - is this an impurity? Please comment.
2. The authors state that the spectra of free base sulfonic acid porphyrin is similar to TPP - is that really correct? Sulfonic acids are very strong acids and should protonate the porphyrin core, which results in significant shifts. Please comment.
3. in line 288, the word "acid" is missing after "sulfonic"
line 334, should read: "PDT relies on the use of..."
line 465, 13C is most certainly not measured at 300 MHz but at 75 MHz - this needs to be corrected also in the Supplementary material.
line 466 the calculated mass and the found mass of the zinc porphyrin sulfonamide need to be switched - the mass spectrum in Fig. S% shows 1105.1817 as the found mass.
Fig. S5,HRMS-ESI of the zinc sulfonamide - this is clearly a very strange spectrum - please comment on the fragmentation and on the purity of the compound! The same holds true for the free base sulfonamide, although to a lesser extent.
Author Response
Reviewer 3
The manuscript "Sulphonamide Porphyrins as Potent Photosensitizers against Multidrug-Resistant Staphylococcus aureus (MRSA): The Role of co-Adjuvants" submitted by Sarabando et al. is an interesting contribution to the field of photodynamic therapy/photodynamic inactivation. The authors present two new compounds, a free base and the corresponding zinc complex of a tetrakis-sulfonamide-substituted porphyrin. They rely heavily on literature procedures for the precursor materials. I think the manuscript is suitable for publication in Molecules after some corrections or clarifications:
- In Fig. 1b, a shoulder appears at the Qy(0,0) transition of the free base sulfonamide porphyrin - is this an impurity? Please comment.
Answer: Fluorescence emission spectroscopy is a very sensitive technique that is able to find emissive impurities in a sample if they are present. We performed the fluorescence emission spectra, and we haven’t found any impurity, as it can be seen in figure 2 c) in the manuscript. As expected for free-base porphyrins, only 2 emission bands have been found. So the presence of the shoulder at Qy(0,0) transition is most likely due to the non-planarity of the porphyrin macrocycle after the introduction of the sulphonamide side chains. This non-planarity changes the molecular orbital energy from HOMO and LUMO increasing the oscillator strength for a “forbidden transition” opening possibility and probability for a weak transition from the fundamental electronic state to the excited state.
- The authors state that the spectra of free base sulfonic acid porphyrin is similar to TPP - is that really correct? Sulfonic acids are very strong acids and should protonate the porphyrin core, which results in significant shifts. Please comment.
Answer: Yes, that is correct. As is shown from the literature, (Ribó, et al. https://doi.org/10.1039/C39940000681 ;see table below) the UV-Vis spectra of protonated porphyrins show 3 Q bands, instead of 4 bands for the non-protonated form ( see table 1 and Figure 1 a) from the manuscript) and a red shift of the Soret band when compared with the non-protonated form. UV-vis spectra of TPPSO3H in non-protonated form and in diprotonated form can be found in Hollingsworth et al (https://doi.org/10.1021/bm201078d) and reproduced in table 1 below.
- in line 288, the word "acid" is missing after "sulfonic"
Answer: The word “acid” was added.
- line 334, should read: "PDT relies on the use of..."
Answer: aPDT is correct. The “a” means antimicrobial.
- line 465, 13C is most certainly not measured at 300 MHz but at 75 MHz - this needs to be corrected also in the Supplementary material.
Thank you for noticing these typos. The frequencies were corrected.
- line 466 the calculated mass and the found mass of the zinc porphyrin sulfonamide need to be switched - the mass spectrum in Fig. S% shows 1105.1817 as the found mass.
Answer: Thank you for noticing this typo. We also inserted the calculated mass and found one in the correct order.
- Fig. S5, HRMS-ESI of the zinc sulfonamide - this is clearly a very strange spectrum - please comment on the fragmentation and on the purity of the compound! The same holds true for the free base sulfonamide, although to a lesser extent.
The compounds were purified by column chromatography and then recrystallization. However, due to the lower solubility in several solvents, we had some difficulty obtaining nice crystals. However, the purity of the products obtained was confirmed by TLC, which showed a unique spot. Moreover, considering the low solubility of TPP(SO2NHEt)4 and the of ZnTPP(SO2NHEt)4 in the solvents allowed in our mass spectrometer their volatilization in the ESI source is also very poor. For that reason, the quality of the HRMS-ESI spectra is low. Some of the peaks are due to the matrix and it is not from our compound. We ask our technician to try several conditions in order to have a better resolved HRMS-ESI spectra but unfortunately, we are not well successful. Unfortunately, we do not have other mass spectrometer ionization sources.
Reviewer 4 Report
The manuscript titled “Sulphonamide Porphyrins as Potent Photosensitizers against Multidrug-Resistant Staphylococcus aureus (MRSA): the Role of co-Adjuvants” is well written and well organized. The document is supported by bibliographic references of academic impact, the methodology and discussion of results are well structured. Therefore I recommend its publication as long as some minor corrections are made.
Some zinc(II) complexes coordinated to arylporphyrins functionalized with sulfonamide groups were reported in 2009 and should be part of the discussion.
· New Porphyrins Bearing Pyridyl Peripheral Groups Linked by Secondary or Tertiary Sulfonamide Groups: Synthesis and Structural Characterization. Manono, Janet; Marzilli, Patricia A.; Fronczek, Frank R.; Marzilli, Luigi G. Inorganic Chemistry (2009), 48(13), 5626-5635
· New Porphyrins Bearing Positively Charged Peripheral Groups Linked by a Sulfonamide Group to meso-Tetraphenylporphyrin: Interactions with Calf Thymus DNA. Manono, Janet; Marzilli, Patricia A.; Marzilli, Luigi G. Inorganic Chemistry (2009), 48(13), 5636-5647
In the introduction, the authors highlight the importance of porphyrins, sulfonamide groups, and co-adjuvants (KI), but they do not indicate the importance of zinc(II). What motivated you to study the coordination complex with this metal? Why zinc(II) and not another transition metal? Describe it on the introduction.
Include in the supplementary information the 1H NMR spectrum of TPP(SO3H)4. Report in the supplementary information the 13C NMR spectrum of TPP(SO2NHEt)4. What are the impurities in the 1H NMR spectrum of TPP(SO2NHEt)4?.
Label each of the proton and carbon positions of the structures of the zinc(II) complex and its ligand, then unambiguously assign the signals in the 13C and 1H NMR spectra.
Include some complementary analytical or spectroscopic characterization technique, such as elemental analysis, IR spectroscopy, etc.
Since the authors do not show X-ray diffraction structures, it would be convenient to determine the lower energy structures of the zinc(II) complex and its ligand using DFT calculations.
Establish in the discussion a comparative analysis of sulfonamide porphyrins with other photosensitizing drugs used against Staphylococcus aureus or other bacteria. Is the presence of zinc(II) relevant? Expand discussion
Author Response
Reviewer 4
The manuscript titled “Sulphonamide Porphyrins as Potent Photosensitizers against Multidrug-Resistant Staphylococcus aureus (MRSA): The Role of co-Adjuvants” is well written and well organized. The document is supported by bibliographic references of academic impact, and the methodology and discussion of results are well structured. Therefore, I recommend its publication as long as some minor corrections are made.
Comment: The authors thank the reviewer for the recognition of the work value and the way it was presented.
- Some zinc(II) complexes coordinated to arylporphyrins functionalized with sulfonamide groups were reported in 2009 and should be part of the discussion.
New Porphyrins Bearing Pyridyl Peripheral Groups Linked by Secondary or Tertiary Sulfonamide Groups: Synthesis and Structural Characterization. Manono, Janet; Marzilli, Patricia A.; Fronczek, Frank R.; Marzilli, Luigi G. Inorganic Chemistry (2009), 48(13), 5626-5635
New Porphyrins Bearing Positively Charged Peripheral Groups Linked by a Sulfonamide Group to meso-Tetraphenylporphyrin: Interactions with Calf Thymus DNA. Manono, Janet; Marzilli, Patricia A.; Marzilli, Luigi G. Inorganic Chemistry (2009), 48(13), 5636-5647
Answer: We thank the reviewer for this alert. We agree that the suggested references are very important and relevant as porphyrins and metalloporphyrins containing sulphonamides with positively charged peripheral N-alkyl pyridinium groups for DNA binding and intercalation. Nevertheless, since the compounds have not been applied as photosensitizers for PDT or aPDT, we thought that the inclusion of those references would be more appropriate in the introduction.
“Porphyrins and metalloporphyrins containing sulphonamides and positively charged peripheral N-alkyl pyridinium groups, have also been prepared and considered for biological applications such as DNA binding and intercalation [42,43].”
- In the introduction, the authors highlight the importance of porphyrins, sulfonamide groups, and co-adjuvants (KI), but they do not indicate the importance of zinc(II). What motivated you to study the coordination complex with this metal? Why zinc(II) and not another transition metal? Describe it on the introduction.
Answer: We thank the reviewer to alert us to this unclear aspect. The following text and references were added in the introduction and discussion sections:
Introduction
“The capability of porphyrin core to complex with a large number of metal cations, giving metalloporphyrins, strongly influences their electronic and photophysical properties by internal heavy-atom effects [53], enhancing the intersystem crossing to the triplet state which is the precursor to the formation of ROS and 1O2 [54].”
Discussion
“It is well-established from the literature [65,67] that metalloporphyrin with closed metal shells, such as Zn(II) is less fluorescent than the corresponding free-base porphyrin, have shorter fluorescence lifetimes, and higher intersystem crossing to triplet states promoted by the spin–orbit coupling mechanism (heavy atom effect) [68].”
- Include in the supplementary information the 1H NMR spectrum of TPP(SO3H)4.
Answer: The 1H NMR spectrum was included as Supplementary Material (figure S6).
- Report in the supplementary information the 13C NMR spectrum of TPP(SO2NHEt)4.
Answer:
The new compounds are TPP(SO2NHEt)4 and ZnTPP(SO2NHEt)4. TPP(SO3H)4 was synthesized by us but it is not new. A sentence was included in the manuscript mentioning the new compounds. TPP(SO2NHEt)4 and ZnTPP(SO2NHEt)4 were characterized by 1H NMR and HRMS as was shown in the manuscript and in Supplementary Material. ZnTPP(SO2NHEt)4 was also characterized by 13C NMR. However, the solubility of TPP(SO2NHEt)4 was very low and it was not possible to obtain 13C NMR spectra with enough intensity to differentiate the carbon signals from the spectrum baseline, even after several attempts and with acquisition experiments of ~48 h. Moreover, since for compound ZnTPP(SO2NHEt)4 the 13C NMR was obtained with acceptable definition and the ZnTPP(SO2NHEt)4 was prepared from TPP(SO2NHEt)4 (complexation with Zinc(II) Acetate) if the 13C NMR of this metallated compound is correct, we must conclude that the one of TPP(SO2NHEt)4 free-base is also correct.
- What are the impurities in the 1H NMR spectrum of TPP(SO2NHEt)4?
Answer: The compound was purified by column chromatography and then recrystallization was attempted to improve the purity. However, due to the lower solubility in several solvents, we have some difficulty in obtaining crystals. However, the product obtained was dried in a vacuum oven for several days and showed a single spot in the TLC control. So, we believe that the principal reason why the 1H NMR spectra are not so clear in the aliphatic region TPP(SO2NHEt)4 is due to the residual signal of deuterated solvent at 2.50 ppm (deuterated-DMSO) and the residual signal of water at 3.33 ppm accompanied by all the sidebands associated with those large signals. In order to clarify the extra signals due to the NMR solvent they were adequately assigned.
- Label each of the proton and carbon positions of the structures of the zinc(II) complex and its ligand, then unambiguously assign the signals in the 13C and 1H NMR spectra.
Answer: The protons were labeled in the manuscript. Additionally, we labeled the protons in the SM figures. As we mention before the solubility of these compounds is quite tricky in the common organic solvents and to have a reasonable 2D (HSQC and HMBC) spectra resolution needed to assign all C-13, it is required a significative acquisition time ( > 24 h) and to use the 500 MHz NMR equipment with cryoprobe available in our lab facilities. Unfortunately, this NMR facility is inoperative for a large period of maintenance (~3 months) and when it was used a higher amount of compound to obtain the NMR spectrum the definition of the signal disappears due to the aggregation of the compound in the NMR tube, which does not allow to identify each carbon unequivocally.
- Include some complementary analytical or spectroscopic characterization technique, such as elemental analysis, IR spectroscopy, etc.
Answer: The studied compounds were fully characterized by modern and non-destructive spectroscopic techniques which allow us to fully characterize them and to establish unequivocally structure. So, considering the complexity of porphyrins the use of IR is important in the characterization of new material based on these macrocycles. Since we have HRMS, the molecular formula of our compounds was confirmed and fortunately using much less amount than the required for Elemental analysis,
- Since the authors do not show X-ray diffraction structures, it would be convenient to determine the lower energy structures of the zinc(II) complex and its ligand using DFT calculations.
Answer: Despite our efforts, it was very difficult to get crystals from these porphyrins. We do agree that DFT studies would be a very interesting suggestion to get more insight from the PS HOMO and LUMO energies. However, what was intended with this paper was an exploratory (proof of concept) considering the potential of porphyrins as PSs and sulphonamides as bacteriostatic compounds. In fact, we decided to take advantage of both chemical entities, by preparing porphyrins containing sulphonamides for bacterial photoinactivation namely against MRSA. Since we get good photoinactivation efficacies with this small “family “of compounds, it is our aim to extend the scope and to prepare porphyrin sulphonamides families that will be interesting through DFT calculation to determine the lower energy structures. We thank the reviewer for this interesting suggestion for future work,
- Establish in the discussion a comparative analysis of sulfonamide porphyrins with other photosensitizing drugs used against Staphylococcus aureus or other bacteria. Is the presence of zinc(II) relevant? Expand discussion
Answer: It was clear and shown in the paper that the Zn Complex ZnTPP(SO2NHEt)4 was more efficient for the inactivation of MRSA when compared with the parent fee base TPP(SO2NHEt)4. In this case, it clearly demonstrated the relevance of Zn cation for the MRSA inactivation outcome. This information was added to the discussion.